# Century Wide Changes in Macronutrient Levels in Indian Mothers’ Milk: A Systematic Review

**DOI:** 10.3390/nu14071395

**Published:** 2022-03-27

**Authors:** Deepti Khanna, Menaka Yalawar, Gaurav Verma, Shavika Gupta

**Affiliations:** 1Independent Researcher, G-914, Jal Vayu Towers, Sector-47, Noida 201304, India; 2Biostatistics and Statistical Programming, Life Sciences-Digital Business Operations, Cognizant Technology Solutions India Pvt Ltd., Manyata Business Park, Nagavara, Bengaluru 560045, India; menaka.shekarappa@abbott.com; 3Abbott Nutrition, Research & Development India, Godrej BKC Plot–C, “G” Block, Bandra Kurla Complex, Bandra East, Mumbai 400051, India; gaurav.verma1@abbott.com; 4Independent Researcher, MP-96, Maurya Enclave, Pitampura, New Delhi 110034, India; shavika@gmail.com

**Keywords:** breastfeeding, human milk, macronutrient, breast milk, lactation, colostrum, transitional milk, mature milk

## Abstract

The purpose of this systematic review was to understand Indian mothers’ milk composition and report changes in it over the past 100 years. A review was conducted in accordance with PRISMA and registered with PROSPERO (CRD42022299224). All records published between 1921 and 2021 were identified by searching databases Google Scholar, ResearchGate, PubMed, and the Cochrane Database of Systematic Reviews. All observational, interventional, or supplementation studies reporting macronutrients (protein, fat, lactose) in milk of Indian mothers, delivering term infants, were included. Publications on micronutrients, preterm, and methods were excluded. Milk was categorized into colostrum, transitional, and mature. In all, 111 records were identified, of which 34 were included in the final review. Fat ranged from 1.83 to 4.49 g/100 mL, 2.6 to 5.59 g/100 mL, and 2.77 to 4.78 g/100 mL in colostrum, transitional, and mature milk, respectively. The protein was higher in colostrum (1.54 to 8.36 g/100mL) as compared to transitional (1.08 to 2.38 g/100 mL) and mature milk (0.87 to 2.33 g/100 mL). Lactose was lower in colostrum (4.5–6.47 g/100 mL) as compared to transitional (4.8–7.37 g/100 mL) and mature milk ranges (6.78–7.7 g/100 mL). The older studies (1950–1980) reported higher fat and protein in colostrum as compared to subsequent time points. There were variations in maternal nutritional status, diet, socioeconomic status, and regions along with study design specific differences of time or methods of milk sampling and analysis. Additionally, advancements in methods over time make it challenging to interpret time trends. The need for conducting well-designed, multicentric studies on nutrient composition of Indian mother’s milk using standardized methods of sampling and estimation for understanding the role of various associated factors cannot be undermined.

## 1. Introduction

Rapid urbanization, increasing food availability, and a sedentary lifestyle are a part of a nutrition transition India is undergoing, along with facing a triple burden of malnutrition. Nearly 24% of women in the reproductive age-group are overweight, 18.7% are underweight, and 57% are anaemic in India [1]. Maternal malnutrition is associated with adverse birth outcomes and increased maternal morbidity [2]. The neonatal mortality rate in India is 23 deaths per 1000 live births according to Sample Registration System, 2018 [3], and India represents 19% of worldwide under-five deaths and 24% of all neonatal deaths [4]. Neonatal mortality continues to remain the largest contributor to all under five deaths in 2020 (47%) which is higher than 1990 estimates (40%) [5]. Geographic and economic disparities increase the child mortality; and the achievement of Sustainable Development Goal (12 per 1000 live births) seems far-fetched for India until it increases its efforts enormously [6].

Early initiation and exclusive breastfeeding significantly reduce neonatal mortality [7,8]. Human milk is nutritionally superior to other liquids and solid foods for infants [9,10]. It is a biological fluid that contains nutritive and non-nutritive components [11,12,13] which are essential for optimal growth and development of infants [14,15,16,17,18,19]. It contains bioactive compounds that help in developing the immune system and intestinal microbiota [20,21], thus providing protection against gastrointestinal, and respiratory infections [22,23] along with allergic diseases [24,25,26]. Breastfeeding is also associated with a reduced long-term risk of inflammatory bowel disease (IBD) [27,28], obesity, adiposity [17,29,30], or diabetes [23,27,31,32].

The composition of human milk is variable and largely depends upon maternal, infant, and environmental factors [17,21,33,34,35]. Physiological factors, including feeding pattern, duration of feeding, and lactational stage are understood well [13,33,36,37]. Conclusive research on the effect of maternal dietary intake, anthropometry, and nutritional status on breast milk composition [17,35,38,39,40,41,42] is ongoing. Infant and maternal genes also contribute significantly to human milk composition [13,36]. Other related factors include maternal education, socio-economic status (SES), ethnicity, and geographical location [43,44,45,46,47,48,49,50,51,52,53]. Understanding breast milk composition becomes very important for a country such as India, which is undergoing a transition along with a pre-existing regional and ethnic diversity.

In India, maternal diets are generally plant-based and may have macro and micronutrient imbalances. This could affect the macronutrient composition of human milk. Studies have shown that with plant-based diets in populations with food insecurity, breast milk fat content may be suboptimal [13,50,54]. Earlier studies from India had also reported the effect of maternal diets and nutritional status on macronutrient content in breast milk [55,56,57,58]. Studies have reported that milk composition of mildly undernourished women is comparable to well-nourished women, possibly due to some metabolic adaptation [44].

It becomes imperative to understand human milk composition and the various factors associated with its composition to formulate more comprehensive guidelines for the promotion of infant care practices during the first 1000 days. This is important for optimum growth and development of infants, irrespective of socio-economic status and regional variation. The work in the Indian subcontinent is limited on composition of human milk. To the best of our knowledge, to date, there has been no systematic review to assess the changes in breast milk composition of Indian mothers. An effort has been made to compile all the published studies on human milk composition of mothers of Indian origin in the last century. In addition, this review intends to critique the evidence of associated data on maternal nutritional status, their diet, the SES, and gestational age. 

## 2. Materials and Methods

### 2.1. Search Strategy and Selection Criteria

A systematic review with an assessment of heterogenous trials in accordance with protocols of Systematic Review and Meta-Analysis (PRISMA) 2020 guidelines was conducted [59]. This review is registered with PROSPERO (CRD42022299224). In the first round, Google Scholar, ResearchGate, PubMed, and the Cochrane Database of Systematic Reviews databases were searched till 30 November 2021 using MeSH key words: ‘breast milk India’, ‘human milk India’, ‘human milk composition India’, ‘breast milk nutrient content India’, ‘Nutrients Indian mothers milk’, ‘Nutrition lactation India’ with Boolean operations ‘AND’ or ‘OR’ for including macronutrients, micronutrients, chemical, nutrient composition publications reported in the past 100 years (1921–2021). Following this, databases were searched using individual nutrient keywords: ‘protein Indian mothers’ milk’, ‘protein breast milk India, ‘fat Indian mothers’ milk’, ‘fat breast milk India’, ‘lactose Indian mothers’ milk’, and ‘lactose breast milk India’, to identify the studies published in the English language. All publications which reported human milk composition of Indian mothers or those reporting human milk nutrient composition from mothers of Indian ethnic origin were included in the initial screening. The search was not limited to abstract and title; all records reporting macronutrient composition were included for full-text screening. None of the study designs were eliminated if they met the search criteria. As no search strategy can guarantee completeness, manual searches were conducted from the list of references of published studies and cross-referencing was performed to identify additional published studies. Lactational stages were categorized as colostrum (≤5 days postpartum), transitional milk (6–15 days postpartum), and mature milk (≥16 days postpartum) [15,60,61]. The studies that did not report any stage of lactation were considered as mature milk basis study details. Studies were considered eligible based on specific inclusion and exclusion criteria:

Inclusion Criteria:All observational, interventional or supplementation studies involving any nutrient in human milk of Indian mothers/in mothers of Indian origin published or accepted in peer reviewed journals were included during initial screening and assessed in detail for eligibility;Only studies which reported any macronutrient (fat, protein, lactose) were included for detailed assessment;Studies conducted on human milk in Indian mothers of term infants were included. Those records which reported both term and preterm mothers were also included but only term data was included in this review;Studies reporting any type of milk (colostrum/transitional/mature) from Indian mothers were included;Studies based on a milk macronutrients composition in maternal disease condition but had a control group as well were included. Data from control group from such publications was referred to for this review;Publications based on conferences, technical reports, Letters to the Editor with study findings, and data reported in Indian Medical Gazette on nutrient composition of Indian mother’s milk were also included.

Exclusion Criteria:Animal and laboratory in vitro studies were excluded;Human milk nutrient composition of mothers of any other nationality other than Indian were excluded;All analytical/methodology papers, review articles, duplicate papers, or the records not reporting macronutrients were excluded;Publications where milk macronutrient composition was only from mothers with a disease condition were excluded.

### 2.2. Screening and Retrieval of Full Text

Screening and eligibility assessment of all 111 studies identified was conducted by two reviewers (DK and GV) based on the title and/abstract (Figure 1). Following this, a total of 22 studies were excluded (review articles (*n* = 14), methodological studies (*n* = 1), studies where human milk was not evaluated (*n* = 2), nutrient data not reported (*n* = 2), full text could not be retrieved (*n* = 3)). An eligibility assessment was conducted for remaining 89 publications. Further, based upon detailed reading, 55 records were excluded owing to either being on only preterm mothers (*n* = 3), reporting data on nutrients other than protein/fat/carbohydrates (*n* = 47) or based on analytical methods (*n* = 2), duplicate papers (*n* = 2), and data only from mothers with disease condition (1). All studies were revisited by SG for identifying any other relevant studies and validating search. Hence, a total of 34 records have been included in this systematic review. The units reported by all the above studies was either g/100 mL or g/100 g or% of nutrient in human milk. Since these units are comparable, all 34 records were included in the review (Appendix A) [43,49,55,58,62,63,64,65,66,67,68,69,70,71,72,73,74,75,76,77,78,79,80,81,82,83,84,85,86,87,88,89,90]. One article published in 1958 reported data from two studies [43]. Data from both studies were considered and treated as P1 (*n* = 6) and P2 (*n* = 40). Control group data (with no disease condition) was included from 4 studies which reported milk macronutrients from mothers with a disease condition (Beri-Beri [63], with children suffering with kwashiorkor [64], pregnancy induced complication [86], anaemia [90]). Supplementation studies that included non-supplemented control groups were also included [74]. Studies which reported both term and preterm mothers milk macronutrients were included but only term mothers’ milk was considered for inclusion for this review [85,87]. 

### 2.3. Data Extraction

During initial screening, the 34 records were examined by one reviewer (DK) to assess and extract data on first author, year of publication, sample size, geographical location, type of milk, macronutrients included in sample collection, method of nutrient estimation and key findings. All studies were further categorized on human milk data available for colostrum, transitional milk, and mature milk. SG and GV participated during detailed screening of all records and any differences between reviewers were discussed and consensus for eligibility and inclusion was made.

### 2.4. Quality Assessment 

The quality of 34 included records were assessed using study quality assessment tools by National Heart, Lung, and Blood Institute (available at: https://www.nhlbi.nih.gov/health-topics/study-quality-assessment-tools (last accessed on 1 February 2022) for systematic evidence of records [91] by DK and MY. In case of any disagreement, it was resolved by discussion. Three different tools were used for cross-sectional and cohort, case series and case–control studies. There were 22 cross-sectional, four cohort, six case–control, and two case series studies reviewed. The quality score for majority of studies (*n* = 29) was fair. Two studies were rated good [73,87] and the score for four studies was poor [62,76,77,78]. Study population and objectives were not clearly defined, and the reliability of data was questionable for these poorly scored records. It was observed that none of the studies included sample size calculation methods and justification for the included samples. Outcomes assessed were not blinded in any of the studies, given the nature of the study. A quality assessment table is included in the Appendix A.

### 2.5. Statistical Analysis 

We could not conduct a meta-analysis as the included studies were very old and diverse. Mean, standard deviation, standard error, and ranges of macronutrients levels in milk as mentioned in the published articles for various stages of lactation and outcome data on associations between macronutrient composition of milk and maternal and infant factors were extracted. Where more than one group mean was available, weighted mean was calculated considering the weights as the size of each group. Weighted mean was calculated, and the graphs were produced in Microsoft Excel. Patterns of change over time were evaluated using structured tables and graphs.

## 3. Results

A total of 111 records were identified, of which 89 were screened for eligibility (Figure 1). In total, 34 (*n* = 34) records reported macronutrients. For ease in reporting, the number of publications is presented decade wise (Figure 2). In the 1920s, no publication was found on Indian mothers’ milk composition. The first and oldest record was from the Indian Medical Gazette published in 1931 [62]. Only one record was obtained in the 1940s [63]. The following four decades were research-intensive with a majority of the work reported on nutrient composition of Indian mothers’ milk. A big contributor was the National Institute of Nutrition (NIN), earlier known as the Nutrition Research Laboratory (NRL). The next two decades (1990s and 2000s) were relatively lean, and the last decade (2010+) seeing a slight increase in publications. Thus, valid assessments over time and across studies could be mostly possible from 1950s onwards.

The maximum number of records were reported from Southern (*n* = 11) and Western regions of India (*n* = 10) followed by Northern (*n* = 9), Eastern region (*n* = 2), and Central India (*n* = 1). There was one record from Burma [62]. The sample size of studies varied from 3 to 232. There were 10 studies that had sample size < 30, 11 studies with sample size between 30 to ≤60, 7 studies between 61 to 100, and 5 studies with sample size > 100. Sample size for one publication was not available [79].

In majority of records (*n* = 20, 58.8%), milk sampling was completed in the morning; only two studies collected samples (5.9%) in the afternoon. The remaining records (35.3%) did not mention the time of milk collection (Appendix A). Only four studies (11.8%) reported whether foremilk [68,74,85] or hind milk [49] was collected. A total of 15 records reported data on colostrum, 11 on transitional, and 24 on mature milk for one or more nutrients. Data from three studies [65,75,81] that did not mention the stage of lactation were considered as mature milk, basis publication details. Majority of the studies were conducted on mothers belonging to low SES (*n* = 14, 41.2%) and two were on high SES [72,84]. A total of eight records compared human milk composition of mothers belonging to one or more SES [49,58,65,69,70,75,81,82]. Remaining records (*n* = 10) did not mention the SES. Only 13 records mentioned the nutritional status of mothers, of which six used Body Mass Index (BMI) for classification [49,58,83,84,88,89]. Effect of maternal diet (type of diet/dietary intake) on macronutrient composition of milk was reported by 10 records [65,68,69,70,71,72,73,74,76,81]. Therefore, maternal health, dietary intake, or nutritional status data were reported by limited records.

A total of 24 studies on fat/fatty acid profile were identified but, after screening, three were excluded. Out of these, two reported only triglyceride levels [88,89] and one reported the fatty acid profile [58]. Finally, 21 studies with total milk fat content were included. The fat content ranged from 1.83 to 4.49 g/100 mL in colostrum, 2.6 to 5.59 g/100 mL in transitional milk and 2.77 to 4.78 g/100 mL in mature milk (Table 1). 

Total of five publications have reported fat levels in colostrum (Figure 3a) [69,80,85,86,87]. Colostrum fat was highest (3.92 g/100 mL) in 1959 [69] in eastern India from women belonging to low SES. Another study conducted nearly 20 years later in 1981 [80] in western India reported similar fat (3.8 g/100 mL) in colostrum. Both studies used different methods of fat estimation (Appendix A). However, three studies conducted beyond 1990s reported a lower fat in colostrum. These studies were conducted in women residing in northern India [85,86,87].

There were five studies that reported fat for at least two stages of lactation (Figure 4a). It was observed that, in early publications (1959 [69] and 1981 [80]), the fat content was higher in colostrum as compared to other stages of lactation, but after that, fat content of colostrum was lower than that in transitional and mature milk [85,87]. This may be owing to advancements in fat estimation and milk sampling techniques over time. Additionally, time of milk collection was not comparable across the studies as it was either in the morning [55] or a mixed sample from morning feed [69], 24 h milk sample [86] or was not reported [80,85,87] (Figure 3a and Figure 4a).

Transitional milk fat was reported by seven publications (Table 1) [55,80,82,84,85,87,90]. Of these, three studies [55,84,90] collected early transitional milk (3rd–11th day), three collected milk between the 6th and 15th day pp [80,85,87] and one study had a wide transitional milk data collection (5th–25th day pp) [82]. The values were higher in early transitional milk (4.48–5.59 g/100 mL) as compared to other studies (2.6–3.08 g/100 mL) reporting late transitional milk. However, all these studies used different analytical methods for estimation (Appendix A).

As compared to colostrum and transitional milk, more publications (*n* = 17) have reported mature milk fat (Appendix A). A majority of the studies (*n* = 12) reported fat between 2.7 and 3.74 g/100 mL [43,55,62,63,66,69,73,74,79,80,85,87], but there were five studies which reported higher levels (4.57–4.78 g/100 mL). Out of these five, four were older [43,65,70,71] and one was a recent study [49]. The study reporting the highest levels (4.78 g/100 mL) was conducted on six South Indian mothers during late lactation (66–100th weeks) [43]. The other three studies reporting higher fat levels were from Western Indian mothers, belonging to different SES with dietary fat ranging from 8 to 117 g/day [55,70,71]. The recent study from Eastern Indian high SES women (primarily fish eating/non-vegetarian) also reported higher levels of fat in hind mature milk (4.57 g/100 mL) [49]. In comparison most (*n* = 12) studies reporting similar ranges were mainly conducted on mothers belonging to poor SES. All 17 studies collected milk sample at different time points and used different methods of estimation (Appendix A).

From a total of 31records reporting protein content in milk in the past 100 years, 14 were in colostrum, 11 in transitional, and 24 in mature milk. Protein content was reported in colostrum ranging from 1.54 to 8.36 g/100 mL, 1.08 to 2.38 g/100 mL in transitional milk, and 0.87 to 2.33 g/100 mL in mature milk (Table 1). Out of a total of 14, four studies collected milk in first three days of birth [55,58,72,86] and reported higher protein (4.58–8.36 g/100 mL) The remaining ten studies had similar protein values ranging from 1.42 to 2.5 g/100 mL (Figure 3b).

The protein in transitional milk was reported highest in 1959 (2.3 g/100 mL) in early milk (3rd–10th day pp) collected from low SES South Indian mothers [55] (Figure 3c). Similar values were reported by Ashdhir in 1962 (2.38 g/100 mL) on milk collected on 8th day from North Indian mothers belonging to high SES [72]. Both studies used Kjeldahl method for estimation. The studies conducted thereafter reported lower values of protein in transitional milk ranging from 1.08 to 1.92 g/100 mL. All these transitional milk studies collected sample between the 5th and 15th day pp, were mostly from women who belonged to low SES, resided in different regions of the country and used different methods of estimation. A majority of earlier studies used Kjeldahl [55,72,78,80,82,83] whereas latter studies used Lowry method [84,85] and milk analyzer [87,90]. Along with differences in SES and analytical methods, the phase of milk collection within the same lactational stage may be an important factor influencing nutrient composition.

The protein content in mature milk was reported highest in earliest studies in the 1930s [62] and 1940s [63]. There was only one study in each of these early decades making it difficult to draw interpretations over time. It was observed that all studies post that reported similar levels for protein (0.87–1.63 g/100 mL) in mature milk (Table 1). Additionally, it was observed that there were regional and SES variations, differences in milk estimation techniques and milk sampling period ranging from 11 days to 2 years. There were 10 studies that reported protein for at least two or more stages of lactation (Figure 4b). Protein content is reported to be the highest in colostrum as compared to other postnatal stages of lactation [55,69,72,75,77,78,80,83,85,87] irrespective of variations in samples and methods of estimation.

The major carbohydrate in human milk is lactose and it is also the most measured carbohydrate in the studies published on Indian mothers in the last century. We included 19 records reporting lactose content (Table 1). A total of 6 studies reported lactose in colostrum, 8 in transitional, and 12 in mature milk. The lactose content in breast milk ranged 4.5–6.47 g/100 mL in colostrum, 4.8–7.37 g/100 mL in transitional, and 6.78–7.7 g/100 mL in mature milk. There were five studies [55,69,72,85,87] that reported lactose for at least two or more stages of lactation (Figure 4c). Changes in lactose levels over lactational stages indicate lactose levels in mature and transitional milk have been higher than colostrum levels across all time points for the past six decades. Although, all studies used different methods of estimation (Appendix A), lactose in mature milk was comparable over time.

## 4. Discussion

Human milk typically constitutes 7% carbohydrates (mostly lactose), up to 5% fat, 0.9% proteins, and lesser quantities of vitamins and minerals [92]. It is reported that lipid content changes with stages of lactation [38]. In this review, fat ranged from 1.83 to 4.49 g/100 mL in colostrum, 2.6 to 5.59 g/100 mL in transitional milk and 2.77 to 4.78 g/100 mL in mature milk. Protein widely varied ranging from 1.54 to 8.36 g/100 mL, 1.08 to 2.38 g/100 mL, and 0.87 to 2.33 g/100 mL in colostrum, transitional, and mature milk, respectively. The lactose content ranged 4.5–6.47 g/100 mL, 4.8–7.37 g/100 mL, and 6.78–7.7 g/100 mL in colostrum, transitional and mature milk, respectively.

The primary objective of this review was to understand the Indian mothers milk composition and report any changes over time. It was noted that older studies from the 1950s–1980s reported higher fat and protein in colostrum as compared to later time points. However, milk is well known to be an extremely variable biological fluid which varies between and within populations, across lactation, within mother during a single nursing and throughout the day [22,93,94,95]. In this review as well, there were inherent variations with regional, SES, and habitual dietary differences along with study design specific differences of varied time and method of milk sampling, and analytical methods. With many studies not reporting sampling, estimation methods, or maternal nutritional status, coupled with advancements in methods over time, it makes it challenging to attribute the observed differences to time trends. It is well known that sampling, storage, and analytical methods play a distinct role in variation in nutrient levels [95] with each method having a different precision, margin of error, sensitivity, and specificity. From the studies included in this review, 11 different methods have been reported for fat estimation, 7 for protein, and 9 for lactose. A total of 17 studies reported milk storage methods, of which 7 froze the samples (either ice–salt mix or −20 degrees) to analyse later and 10 studies analysed on the day of collection. In total, 22 (64.7%) studies reported the timing of milk collection. In majority of records (59%), milk was collected in the morning, two studies collected in the afternoon and remaining records did not mention the time of milk collection (Appendix A). It was observed that fat and protein values were higher in colostrum if sample was taken between 24–48 h after delivery [55,58,86]. Detailed review indicates that owing to lack of consistent and standardized methods of study design, sampling and analysis, the time trends are generally inconclusive.

Apart from methodological factors, human milk is known to be influenced by maternal nutritional status, dietary intake, socioeconomic status, and regional variations [23,35,39,40,41,50,96]. In this review, four studies objectively compared different maternal BMI categories and its potential impact on milk composition [58,83,88,89]. All four studies reported positive association of maternal BMI and protein in colostrum indicating higher protein in colostrum of well-nourished as compared to under nourished mothers. As comparative BMI records were limited, we elucidated individual records which reported maternal nutritional status. There were nine additional records which could be used for this interpretation (two reported BMI [49,84], five reported weight [55,67,68,72,81] and two directly reported the nutritional status [74,78]. This individual records assessment indicated that protein in transitional milk of normal-nourished mothers [72,78,84] was between 1.08 and 2.38 g/100 mL. Similarly, the amount of protein in mature milk of normal nourished mothers was 0.94–1.63 g/100 mL [55,68,72,78,83] but was slightly higher (0.86–1.22 g/100 mL; not significant) than that reported for undernourished mothers [55,74,81,83]. Therefore, basis this review, we observe there may be a mild association of maternal BMI on milk protein (specially in colostrum), but needs to be further confirmed as data on maternal BMI and milk protein composition was limited. A recent review reported mixed results for maternal BMI and milk protein with almost 58% of the studies citing no association [41]. Similarly, another review reported protein concentration in breast milk was comparable between overweight and normal weight women in different stages of lactation [40].

Regarding fat, none of the records compared milk fat from mothers in different BMI categories. Based upon individual records, two studies on undernourished mothers reported lower fat in mature milk [55,74] as compared to a recent study in normal nourished mothers reporting higher fat [49]. Two records that compared lactose in milk of mothers in different BMI categories, reported no statistically significant difference in lactose in colostrum [88,89]. Thus, inadequate data on Indian mother’s milk exploring an association between maternal BMI with fat and lactose limits the ability for reporting conclusive associations. A recent review reported that BMI was associated with higher lactose levels at different stages of lactation [40], whereas another review stated that there was no association between lactose and maternal nutritional status in two-thirds of the studies included in the review [41].

From this review, only three studies assessed the association between maternal haemoglobin and milk macronutrient composition [69,83,90]. Two publications reported no significant effect of maternal haemoglobin on breastmilk macronutrient composition [69,83]. However, a recent study reported that severe, and moderate anaemia causes significant changes in fat, lactose, and protein content of breast milk, while mild anaemia has little effect on lactose and protein contents of breast milk but not on fat content [90].

A majority of the studies in the 1950s [43,64,65,67,68,69] and 1960s [74] conducted on women consuming poor cereal based low protein diets reported a negligible effect on lactational performance and milk composition. Similar observations were made by two recent studies [88,89]. An older study reported that low dietary fat could be one of the reasons for low fat content in milk of Indian mothers when compared to mothers of developed countries [55]. Other studies stated that higher fat content in milk could be due to a customary high fat diet in certain regions of the country [49,58,86,90]. Sinha et al., in 1959 [69], summarized that a moderate variation in diet did not affect the milk composition. It was reported that an increase in dietary protein and fat increased milk protein and fat, respectively, but only up to a certain level, and an increase beyond general sufficiency levels had no additional beneficial effect [43,65,70,71,72,74]. Another study reported that protein supplementation did not change protein concentration in milk but improves the quality of milk protein in the mothers on poor diet [73]. Unlike fat and protein, dietary lactose did not report to have a significant effect on milk lactose levels [70,72]. Thus, basis the review, it may be understood that if dietary protein and fat intakes are adequate, no additional benefit may be observed with additional supplementation.

Two studies reporting type of diet (vegetarian, lactovegetarian, non-vegetarians) had no significant difference in protein content of breast milk [76,81]. However, Mukherjee and Anwikar in 1959 [68] reported that the milk protein may be slightly higher in mothers on mixed diets (1.34 g/100 mL) as compared to those on complete vegetarian diet (1.29 g/100 mL). Thus, it was observed that type of diet (vegetarian or non-vegetarian) either did not report or had only a slight association with milk macronutrients, specifically protein.

In this review, most of the studies *(n* = 14) were conducted on women belonging to lower income group with eight studies compared data from different income groups. Studies conducted in late 1950s [65,69,70] demonstrated that breast milk of women belonging to very poor SES contained significantly less fat than those from middle- and higher-income group. The studies from other parts of the country also reported the same in the later years [49,82]. However, some studies reported that fat, protein, and lactose in breast milk of poor Indian women was comparable with the reported values from other developed countries which had better diets and socioeconomic status indicating these factors did not have a significant effect on macronutrient content of breast milk [55,75]. It is noteworthy that every nutrient in breastmilk evolves during the course of lactation to meet the nutrient requirements of growing infant. These studies have discussed that these similarities could be due to some metabolic adaptation but may lead to maternal deprivation of nutrients. This interpretation needs to a validated basis a more recent and standardized data on dietary intakes.

It becomes important to understand the entire human milk biological system for better implication of this for vulnerable segment of the population and to cover women belonging to very poor socio-economic status, different nutritional status, residing in different geographical locations with regional and ethnic differences [60,97]. The future studies need to be designed in such a way that they guide policy makers and regulatory bodies for micro-level planning and selective targeting for better implementation and adherence to Infant and Young Child Feeding guidelines [10].

### Strengths and Limitations of this Review

India is historically a natural promoter of breastfeeding. This is the first ever century based systematic review published on Indian mothers’ milk macronutrients composition. With high rates of infant and neonatal mortality, along with a triple burden of malnutrition, this review is a timely compilation of milk macronutrient composition across time, from different regions and states of India, across stages of lactation, SES, maternal nutritional status, and diets. 

The limitation of this review was the large heterogeneity in sampling and estimation along with limited records focussing on the impact of maternal diets, nutritional status, or regional differences on the macronutrient composition on milk. This impeded our ability to not only conduct a meta-analysis but also draw time trends. This limitation surely highlights the future need for conducting well designed, multicentric studies using standardized methods on Indian mothers’ milk composition.

## 5. Conclusions

This is the first systematic review studying changes in macronutrient composition of Indian mothers’ milk over the past century and reporting the role of multiple factors impacting it over time. There are individual records with data from mothers belonging to different regions, socio-economic strata along with variations in gestational age, lactational stage, maternal dietary intake, and nutritional status. The studies prior to the 1980s reported the highest fat in the colostrum (vs. other stages of lactation) than that reported by recent studies. Protein was the highest in colostrum as compared to other postnatal stages of lactation in all the studies. Along with other methodological differences, the phase of milk collection within the same lactational stage may be an important factor. With India undergoing a socio-economic, demographic, and nutritional transition, along with recent advancements in methodological techniques (milk sampling, handling, storage, estimation), more standardized research in this area is necessary. A similar understanding for micronutrient composition of human milk is desirable considering the micronutrient deficiency prevalence.

## Figures and Tables

**Figure 1 nutrients-14-01395-f001:**
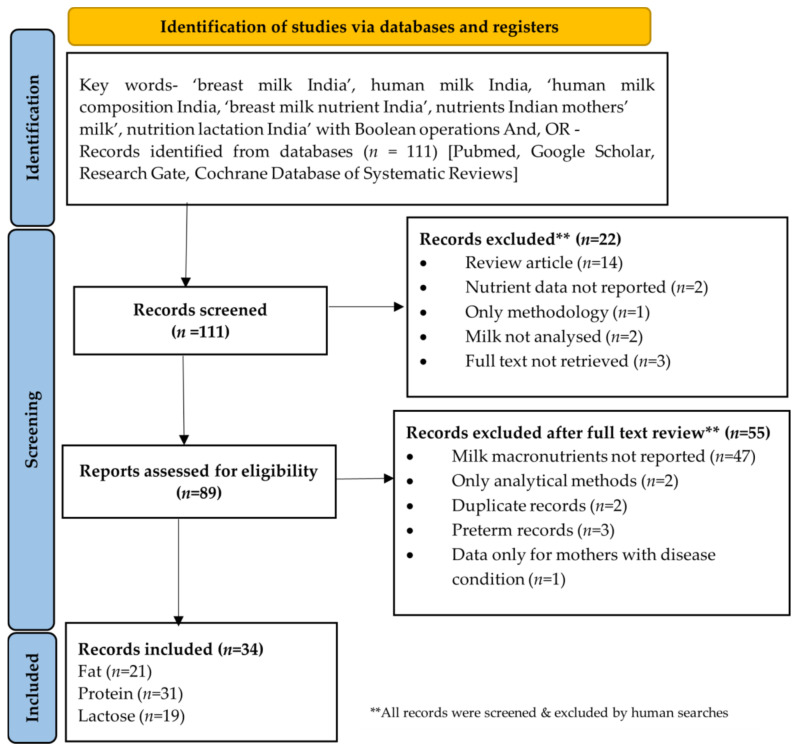
PRISMA flow chart for systematic review study inclusion.

**Figure 2 nutrients-14-01395-f002:**
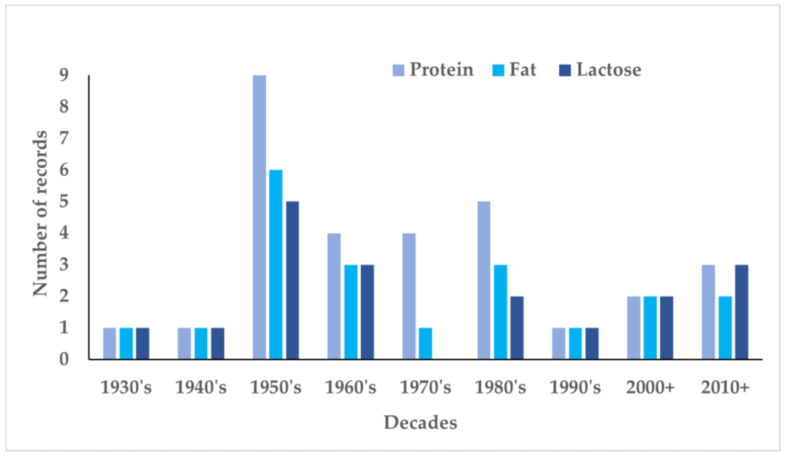
Decade wise records on protein, fat and lactose in Indian mothers’ milk (1930–2010+).

**Figure 3 nutrients-14-01395-f003:**
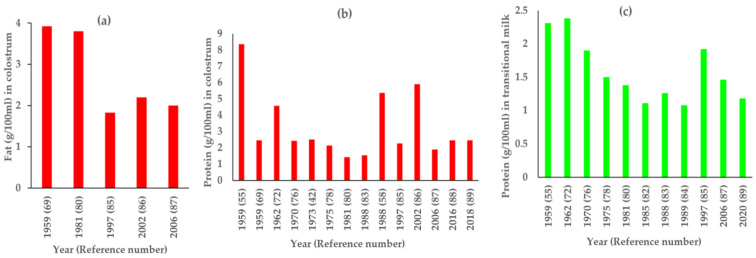
Milk fat and protein (g/100mL) over time (**a**) colostrum fat, (**b**) colostrum protein, and (**c**) transitional protein.

**Figure 4 nutrients-14-01395-f004:**
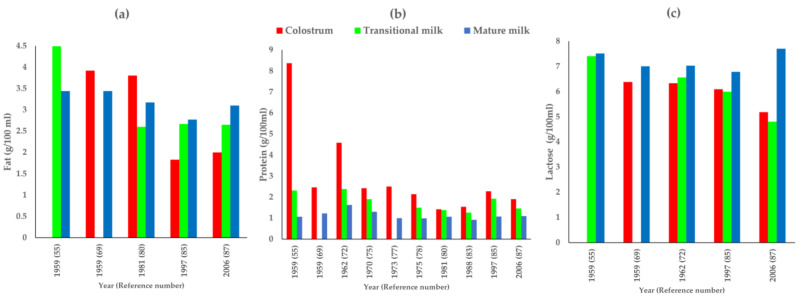
Studies reporting (**a**) fat (g/100 mL), (**b**) protein (g/100 mL) and (**c**) lactose (g/100 mL) for at least two stages of lactation.

**Table 1 nutrients-14-01395-t001:** Details of macronutrient content in colostrum, transitional and mature milk by various records (*n* = 34).

S.No.	First Author, Year (Total N)	Fat (g/100 mL)			Protein (g/100 mL)			Lactose (g/100 mL)		
		Colostrum	Transitional	Mature	Colostrum	Transitional	Mature	Colostrum	Transitional	Mature
1	Bunce, 1931 [62] (N = 8)			3.23 † (*n* = 3)			2.33 † (*n* = 3)			6.78 † (*n* = 3)
2	Sundararajan, 1941 [63] (N = 40)			3.3 † (infants < 4 m–1.1–5.9, *n* = 17; infants > 4 m-1.0–6.9, *n* = 23)			2.05 † (infants < 4 m–1.4–3.4, *n* = 15; infants > 4 m–0.5–3.6, *n* = 20)			7.26 † (infants < 4 m–3.2–9.5, *n* = 21; infants > 4 m-4.4–9.0, *n* = 19)
3	Srinivasan, 1954 [64] (N = 31)						1.27 * (0.89–1.80, *n* = 25)			
4	Karmarkar, 1958 [65] (N = 175)			4.68 † (*n* = 175)						
5	Gopalan P1, 1958 [43] (N = 6)			4.78 (*n* = 6)			1.19 † (*n* = 6)			6.99 † (*n* = 6)
6	Gopalan P2, 1958 [43] (N = 40)			3.34 ± 0.242 * (*n* = 40)			1.06 ± 0.036 * (*n* = 40)			7.47 ± 0.072 * (*n* = 40)
7	Belavady, 1959 [66] (N = 29)						0.96 † (*n* = 29)			
8	Belavady, 1959 [55] (N = 191)		4.49 ± 0.47 * (*n* = 9)	3.44 † (*n* = 109)	8.36 ± 0.921 * (*n* = 18)	2.31 ± 0.435 * (*n* = 74)	1.07 † (*n* = 146)		7.41 ± 0.161 *	7.51 † (*n* = 146)
9	Belavady, 1959 [67] (N = 36)			2.85 † (*n* = 20)			1.05 † (*n* = 36)			7.67 † (*n* = 20)
10	Mukherji, 1959 [68] (N = 35)						1.24 † (0–1 year- 0.7–2.8, *n* = 25; 1–2 year-0.8–2.3, *n* = 10)			
11	Sinha, 1959 [69] (N = 63)	3.92 * (1.6–5.8, *n* = 9)		3.44 ± 1.50 * (1.0–7.0, *n* = 54)	2.46 * (1.83–3.84, *n* = 9)		1.22 ± 0.28 * (0.84–2.3, *n* = 54)	6.38 * (5.82–7.0, *n* = 9)		7 ± 0.17 * (67–7.6, *n* = 54)
12	Karmarkar, 1959 [70] (N = 232)			4.6 † (*n* = 232)			1.33 † (*n* = 232)			7.17 † (*n* = 232)
13	Karmarkar, 1960 [71] (N = 60)			4.49 † (*n* = 60)			1.37 † (*n* = 60)			7.15 † (*n* = 60)
14	Ashdhir, 1962 [72] (N = 10)				4.58 ± 0.83 * (3.31–5.88, *n* = 10)	2.38 ± 0.54 * (1.05–3.80, *n* = 10)	1.63 ± 0.4 * (1.0–2.14, *n* = 10)	6.33 ± 0.45 * (5.61–7.18, *n* = 10)	6.56 ± 0.68 * (5.59–7.78, *n* = 10)	7.03 ± 0.71 * (5.71–7.86, *n* = 10)
15	Deb, 1962 [73] (N = 20)			2.99 * (2.4–3.8, *n* = 20)			0.88 * (0.70–1.18, *n* = 20)			6.88 * (6.8–7.1, *n* = 20)
16	Karmarkar, 1963 [74] (N = 60)			3.55 * (*n* = 5)			1.08 * (*n* = 5)			
17	Khurana, 1970 [75] (N = 194)				2.425 ± 0.510 * (0.857–4.237, *n* = 28)	1.904 ± 0.6 * (0.811–3.400, *n* = 30)	1.3 † (0.250–3.600, *n* = 136)			
18	Jathar, 1970 [76] (N = 48)						1.58 ± 0.06 * (*n* = 48)			
19	Rao, 1973 [77] (N = 31)				2.5 ± 0.54 * (*n* = 8)		1 † (*n* = 24)			
20	Agarwal, 1975 [78] (N = 97)				2.14 ± 0.98 * (0.89–4.12, *n* = 21)	1.5 ± 0.69 * (0.81–2.91, *n* = 17)	0.99 ± 0.41 * (0.88–3.12, *n* = 59)			
21	Belavady, 1978 [79] (N = NA)			3.23 †						
22	Rao, 1981 [80] (N = 70)	3.8 ± 1.28 * (*n* = 4)	2.6 ± 1.27 * (*n* = 12)	3.17 † (*n* = 52)	1.42 ± 0.071 * (*n* = 3)	1.38 ± 0.0537 * (*n* = 7)	1.07 † (*n* = 34)			
23	Bijur, 1985 [81] (N = 50)						1.22 † (*n* = 50)			
24	Kumbhat, 1985 [82] (N = 50)		3.079 † (*n* = 25)		1.34 * (*n* = 25)	1.27 † (*n* = 25)	1.235 † (*n* = 25)		6.9 † (*n* = 25)	
25	Raghuvanshi, 1988 [83] (N = 121)				1.54 ± 1.16 * (*n* = 10)	1.26 ± 0.17 * (*n* = 7)	0.922 † (*n* = 237)			
26	Garg, 1988 [58] (N = 35)				5.36 † (WN- 4.5–6.8, *n* = 20; UN- 2.6–6.8, *n* = 15)					
27	Patil, 1989 [84] (N = 54)		4.48 ± 1.5 * (1.54–9.0, *n* = 54)			1.08 ± 0.42 * (0.70–3.44, *n* = 54)			6.51 ± 1.28 * (4.0–9.4, *n* = 54)	
28	Paul, 1997 [85] (N = 52)	1.83 ± 0.58 * (*n* = 23)	2.67 † (*n* = 23)	2.77 ± 0.80 * (*n* = 23)	2.27 ± 0.93 * (*n* = 23)	1.92 † (*n* = 23)	1.08 ± 0.63 * (*n* = 23)	6.09 ± 1.49 * (*n* = 23)	5.99 † (*n* = 23)	6.78 ± 1.75 * (*n* = 23)
29	Kaushik, 2002 [86] (N = 80)	2.2 ± 0.8 * (*n* = 20)			5.9 ± 0.9 * (*n* = 20)			4.5 ± 0.9 * (*n* = 20)		
30	Narang, 2006 [87] (N = 86)	2.0 ± 0.58 * (*n* = 41)	2.65 † (*n* = 41)	3.1 ± 0.69 * (*n* = 41)	1.9 ± 0.69 * (*n* = 41)	1.46 † (*n* = 41)	1.1 ± 0.48 * (*n* = 41)	5.18 ± 0.44 * (*n* = 41)	4.8 † (*n* = 41)	7.7 ± 1.3 * (*n* = 41)
31	Roy, 2013 [49] (N = 217)			4.57 † (*n* = 209)						
32	Dias, 2016 [88] (N = 63)				2.46 ± 1.25 * (0.6–7.1, *n* = 63)			6.47 ± 1.45 * (4.2–9.7, *n* = 63)		
33	Kothari, 2018 [89] (N = 63)				2.46 ± 1.25 * (0.6–7.1, *n* = 63)			6.47 ± 1.45 * (4.2–9.7, *n* = 63)		
34	Divedi, 2020 [90] (N = 132)		5.59 ± 0.54 * (*n* = 66)			1.18 ± 0.15 * (*n* = 66)			5.63 ± 0.41 * (*n* = 66)	

Colostrum: ≤5 days postpartum; Transitional milk: 6–15 days postpartum; Mature milk: ≥16th day postpartum; * Values are reported mean and SD from the records; † Values are calculated as weighted mean of different groups reported in the records. Values in parentheses include range, N = total study sample size; *n* = sample included; NA: Not available, WN: Well nourished, UN: Undernourished, m: months.

## Data Availability

All published studies were majorly accessed from various databases and retrieved from journal libraries. All studies including excluded studies are available with Abbott and can be accessed with the permission of Abbott Nutrition. Requests can be sent to deekay11in@gmail.com.

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
