# Peer review of "Century Wide Changes in Macronutrient Levels in Indian Mothers’ Milk: A Systematic Review"

_nutrients, 2022, doi:10.3390/nu14071395_

Round 1

Reviewer 1 Report

The authors perform a systematic review to assess secular changes in human milk contents in India over the last 100 years reporting data from 34 studies.

Methods: the protocol was recently registered on PROSPERO and generally is clearly described. Inclusion / exclusion criteria do not state the authors' approach to mothers with diseases, yet para 2.2 implies that such groups were excluded. Please clarify what was the target population group of mothers. Where they to be disease free, representative of the general population, or were all groups eligible?

Methods, Statistics Para 2.5 states: "There was no formal comparison between any groups, so hypothesis testing was not performed". But as shown in Figure 4, the focus of this paper is on changes in fat and protein contents over time which is indeed a comparison. How were the trend lines in Figure 4 calculated? Without formal assessment of change, these trends are not justified.

Results Table 1. Only one paper was found in the 1930's and 1940's, hence it may be that valid comparisons are possible only from 1950's. Furthermore, it would be helpful if Table 1 included an indication of total sample size not only study counts.

Results Line 214-15: "in early publications (1959 [34] & 1981[45]), the fat content was higher in colostrum as compared to other stages of lactation" in contrast to vice versa after that [50,52]. This finding is highly dubious and sheds major doubt on the validity of those early studies, and by consequence on the apparent secular trends in fat and protein which appear to be largely driven by those early studies (Figure 4 and 5).

Results Line 265: in contrast to colostrum and transitional milk, there were no obvious changes over time in macronutrient contents of mature milk, which has the far greater nutritional role. However, Discussion Line 362 describes the huge differences in assay methods used across studies. Hence, even if statistically valid changes were seen with time, it would be impossible to tell if these were real or related to changes in study methodology.

Presentation:

Table 2 spans 3 pages and 'doesn't work' as a table as it is difficult to see which columns the various values belong to.

Figure 5 should be designed with the same orientation as Figure 4 to allow easier comparison.

Author Response

Please check the attachment, thank you.

Reviewer 2 Report

Review of Century wide changes in macronutrient levels in Indian mothers’ milk: A systematic review

Introduction

Lines 38-39 – different styles for “under five”.

Methods

Line 85 – This is but one example of where the English precision can be improved to make the impact of the manuscript more significant. In the review, the summarizing word “like” is used to refer to a list. Do authors use ‘like’ = for example, or does ‘like’ = “all inclusive”? In line 85, authors list key words included in the search – are these all of the key words that were searched, or just an example of some words that were used in the literature searches?

Line 116 – inconsistent line indentation.

Figure 1 – low resolution image in reviewer copy.

Important consideration - Could authors explain why preterm records were excluded?

Results

Figure legends are too brief, the captions do not allow the figures to stand alone.

Table 2 is informative; however, formatting errors persist. Suggestion would be to re-format the table in landscape view.

Discussion

Lines 361-366 – authors briefly speculate on the role of measurement of nutrients over the past 70 years. Could authors provide more insight into precision, margin of error, sensitivity of the analytical tools used in the 20th century versus today, year 2022? Including the equipment, storage of milk, for analysis.  

Overall, a good start, please try to refine, strengthen and clarify all aspects of the manuscript story.

Author Response

Please check the attachment, thank you.

Reviewer 3 Report

Overall

The objective of the review is to summarize the knowledge concerning century wide changes in macronutrient levels in Indian mothers’ milk. The review covers the aim of the journal and the subject investigated is of worldwide interest. However, it would benefit the reader if the authors will addressed some points and add additional information. References should be supplemented and updated. Moreover, the detailed discussion of the methods used for macronutrients determinations among analyzed papers is missing. It should be figure out that in the analyzed period of time, the methods were significantly improved, which had an unquestionable impact on the obtained results.

Introduction

I suggest to support introduction section with a higher number of references. Moreover, for some statements there are no relevant references or references provided are not the latest (see line 44-50 among others).

Line 44-47

“It provides all essential nutrients such as carbohydrates, proteins, enzymes, growth factors, salts, fats, hormones, and anti-infective factors [7,8] which are essential in normal growth and development of infants.” the statement requires clarification

Line 51-52

“The composition of human milk is variable as it depends upon various factors including maternal nutritional status and dietary intake [11]”. – The main influencing factors are not mentioned. Moreover, the statement requires supplementing and updating of references.

In Introduction and Methods sections basic definitions, namely colostrum, transitional and mature milk are missing.

In Results and Discussion sections there is no data concerning the type of milk, namely hind and foremilk. The indication of the type of milk is extremely important especially in the determination of fat concentration.

The main concern

I have serious doubts as to whether the differences in the data presented by the authors on the basis of the analyzed publications actually result from differences in concentration, or rather from the methods used. The methods used more than 50 years ago certainly differ from the methods used nowadays in terms of sensitivity and specificity. This fact should be thoroughly discussed, in my opinion, as the main limitation in this systematic review. Another issue that requires more extensive discussion is the period of the first days of lactation. Comparing the 2-3 day of lactation with the 5th day of lactation, we will observe significant changes in the quantitative composition of colostrum. Moreover, different authors provided different ranges for each stage of lactation, e.g. Belavady, 1959 and Kaushik, 2002 for colostrum and Divedi, 2020 for transitional milk, which should also be emphasized as a limitation when comparing results and drawing conclusions. Failure to take into account these differences may affect the final conclusions.

Limitations must be clearly indicated as a subchapter in Discussion section.

Moreover, the discussion on the potential impact of the number of samples in the analyzed groups and the range of lactation defined for mature milk has not been presented in a comprehensive manner and taking into account the current literature in this area.

Minor

Please, put dots in the numbers in correct position – see Table 2 and throughout the Results section.

Figure 3, 5 and 6 - the description of the x axis is missing.

Author Response

Please check the attachment, thank you.

Round 2

Reviewer 3 Report

Dear Authors,

The quality of the manuscript has been greatly improved.

However, there are two issues that still require attention.

1st - The modified abstract is too long.

2nd - I feel a bit unsatisfied with the conclusions, which in my opinion require direct link to the data analyzed in systematic review.

Author Response

The quality of the manuscript has been greatly improved.

Thank you for acknowledging and deeply reviewing.

However, there are two issues that still require attention.

1st - The modified abstract is too long.

Thanks for indicating. Have modified the abstract. The word limit has now been revised & reduced from 320 to 268.

2nd - I feel a bit unsatisfied with the conclusions, which in my opinion require direct link to the data analyzed in systematic review.

Thanks for noting. We have modified the text to link it well with the systematic review findings.

Additionally, basis overall review, have tried to describe methods and results sections better as well (minor revisions). All changes in track in revised MS. Please see attachment as well.  
